

# Revisiting the synoptic-scale predictability of severe European winter storms using ECMWF ensemble reforecasts

Florian Pantillon, Peter Knippertz, and Ulrich Corsmeier

Institute of Meteorology and Climate Research, Karlsruhe Institute of Technology, Karlsruhe, Germany

*Correspondence to:* Florian Pantillon (florian.pantillon@kit.edu)

**Abstract.**

New insights into the synoptic-scale predictability of 25 severe European winter storms of the 1995–2015 period are obtained using the homogeneous ensemble reforecast dataset from the European Centre for Medium-Range Weather Forecasts. The predictability of the storms is assessed with different metrics including the track and intensity to investigate the storms' dynamics and the Storm Severity Index to estimate the impact of the associated wind gusts. The storms are correctly predicted by the ensemble reforecasts up to 2–4 days ahead only, which restricts the use of ensemble average and spread to short lead times. At longer lead times, the Extreme Forecast Index and Shift of Tails are computed from the deviation of the ensemble reforecasts from the model climate. Based on these indices, the model has some skill in forecasting the area covered by extreme wind gusts up to 10 days, which indicates clear potential for the early warning of storms. However, a large variability is found between the predictability of individual storms and does not appear to be related to the storms' characteristics. This may be due to the limited sample of 25 cases, but also suggests that each severe storm has its own dynamics and sources of forecast uncertainty.

## 1 Introduction

One of the most important natural hazards over Europe arises from winter storms associated with low-pressure systems from the North Atlantic, also referred to as cyclonic windstorms (e.g. Roberts et al., 2014). These storms are therefore the focus of various fields of research involving the weather and climate communities but also the windpower and reinsurance industries. At longer time scales, numerous studies are dedicated to the estimation of the footprint and return period of winter storms and often require a combination of dynamical and statistical models (Della-Marta et al., 2009; Hofherr and Kunz, 2010; Donat et al., 2011; Haas and Pinto, 2012; Seregina et al., 2014). A crucial but disputed question lies in the trends in frequency and intensity of winter storms in the current and future climate, which still differ between climate models and between identification methods (see Feser et al., 2015, for a review). At shorter time scales, most studies concentrate on the detailed investigation of case studies of severe storms and of their forecast in numerical weather prediction systems.

Although the general lifecycle of extratropical cyclones has been described for almost one century, the intensification of the storms and the generation of strong winds – responsible for most of the damages created by the storms – involve physical processes of different scales that are still not fully understood. Recent advances have resulted from the attention drawn by



devastating storms. The damaging winds over southeast England during the "Great Storm" of October 1987, which were observed at the tip of the cloud head bounding the bent-back front, now form the archetypal example of a phenomenon known as the sting jet (Browning, 2004). The destructions caused by storm Lothar over central Europe in December 1999 revealed the importance of diabatic processes in a way similar to a diabatic Rossby wave for the rapid intensification of the storm over

the North Atlantic (Wernli et al., 2002). The severe wind gusts observed during the passage of storm Kyrill in January 2007 over central Europe finally emphasized the role of the convection embedded in the cold front and including the formation of cold-season derechoes (Fink et al., 2009; Gatzen et al., 2011).

These historical storms were poorly forecast when they occurred and thus captured an even larger attention in the weather research community, which resulted in a prolific scientific literature on specific storms. In particular, Buizza and Hollingsworth

(2002) early recognized the potential of the ensemble prediction system of the European Centre for Medium-Range Weather Forecasts (ECMWF) to forecast the storms Anatol, Lothar and Martin in December 1999. They showed that the ensemble forecast offers a more consistent picture between different initialisations than the deterministic forecast and additionally provides early indications of the chance of an intense storm. Lalaurette (2003) further showed that the extremeness of the ensemble forecast, measured by its deviation from the model climate, allows to identify areas of unusually strong winds up to 120 h lead time

in the case of Lothar, although it fails in the case of Martin. Petroliagis and Pinson (2014) and Boisserie et al. (2016) recently extended this methodology to longer periods with either operational or retrospective forecasts. While they found constrasting results from case to case, the authors confirmed the potential of ensemble forecasts for the early warning of severe European storms.

Following these previous studies, the predictability of severe European winter storms is investigated here for a 20-year period

in an ensemble prediction system by taking advantage of the recently available ECMWF retrospective forecast (reforecast). While reforecasts are originally designed for calibrating the operational forecasts, which result in a significant improvement in forecast skill, they also represent a homogeneous dataset that is ideal for comparing historical events (Hamill et al., 2006, 2013). The predictability of severe storms is thus not restricted to single case studies here but encompasses a large number of events that allow a statistical analysis. Three metrics are combined to assess the predictability in regard to different properties

(Figure 1): the dynamics are evaluated with the track and intensity of the storms and the impact is estimated with the strength of wind gusts, while the potential for early warnings is computed from the area of predicted gusts that are unusually strong compared to the model climate. These metrics are further generalized to situations with less severe or no storms to ensure that the results are not biased by the focus on extreme events.

The manuscript is organized as follows. Section 2 describes the reforecast and reanalyses model data and the selection of

severe storms, as well as the 3 different methods that are used to assess the predictability of the storms in the data. Section 3 presents the results obtained for general storm characteristics and using either the ensemble average and spread or individual ensemble members. Section 4 discusses the skill for early warning using either selected storms or the whole dataset. Section 5 finally gives the conclusions of the study.



## 2 Data and methods

### 2.1 Model data

This study extensively makes use of the ensemble reforecast from the ECMWF (Hagedorn et al., 2008, 2012). The ensemble reforecast is based on the current version of the operational model but with a lighter configuration to reduce computing time. It
is initialised from the ERA-Interim reanalysis (Dee et al., 2011) and ensemble members are obtained from initial perturbations computed with singular vectors. In contrast to the operational model, stochastic perturbations of physical processes are not applied to the ensemble members. Since mid-May 2015, the ensemble reforecast contains 10 perturbed members in addition to a control member and it is run twice a week – every Monday and Thursday at 00 UTC – for the current date in the past 20 years. Until mid-March 2016, when the model resolution was upgraded, the horizontal grid spacing was approximately 30 km
for the first 10 days and was then coarser at longer lead times until 46 days. All 10-day ensemble reforecasts computed between mid-October 2015 and mid-March 2016 are used here, which represents a homogeneous dataset of nearly 10,000 individual reforecasts for the winter seasons 1995/96 to 2014/15.

The reforecasts are verified against the ECMWF Retrospective Analysis (ERA)-Interim, which is available since 1979 and is computed with a horizontal grid spacing of approximately 80 km, corresponding to a pre-2006 version of the operational
model (Dee et al., 2011). Variables of interest include the Mean-Sea-Level Pressure (MSLP) output at 00, 06, 12 and 18 UTC and wind gusts output from short-range forecasts initialized at 00 and 12 UTC. Although it has been widely used for climatological studies of winter storms, ERA-Interim has recently been criticized for underestimating the deepening rate of storms and the strength of winds (Hewson and Neu, 2015). In particular, the low horizontal resolution of ERA-Interim is not sufficient at representing the mesoscale structure of the storms. The next generation of ECMWF reanalysis ERA5 may alleviate
this limitation thanks to its horizontal grid spacing of about 30 km but it is still under development. As the focus here is on the synoptic-scale aspects of winter storms, ERA-Interim is used as a reference, while caution is taken for the interpretation of strong winds that could be related to mesoscale structures.

Significant historical storms are selected to investigate their predictability in the ensemble reforecast. The selection is made using the "XWS open access catalogue of extreme European windstorms" provided by Roberts et al. (2014), which contains
the 50 most severe storms for the 1979–2012 period. The catalogue is based on ERA-Interim dynamically downscaled with the Met Office Unified Model and recalibrated with observations. It is available online at http://www.europeanwindstorms.org/ and was updated with two additional storms for the winter season 2013/14. Following the time period of the ensemble reforecast, the storms that occurred between mid-October and mid-March from 1995/96 to 2014/15 are selected here. One storm occurring in late March is excluded through the restriction to the winter period, expecting that the mid-October to mid-March time span
of the reforecast is relevant for severe storms. The selection results in the 25 storms listed in Table 1. The storm names are those given by the Free University off Berlin when available, with alternative names in brackets when relevant. They were completed for a few storms with respect to the original catalogue of Roberts et al. (2014).



## 2.2 Storm tracking

The 25 selected storms are tracked both in ERA-Interim and in the members of the ensemble reforecast, using the algorithm described by Pinto et al. (2005) and originally developed by Murray and Simmonds (1991). In a first step, maxima are identified in the Laplacian of MSLP interpolated on a polar stereographic grid then minima in MSLP are looked for in their vicinity. The

Laplacian of MSLP is closely related to the quasi-geostrophic vorticity; thus the algorithm is similar to tracking maxima in low-level vorticity. In a second step, the mimima of MSLP are connected between subsequent model outputs every 6 h to form tracks, if their displacement velocity remains consistent in time. As the focus is on severe storms here, the obtained tracks are filtered to exclude storms with a weak Laplacian of MSLP or with a duration of less than 24 h. However, the algorithm is applied hemisphere-wide and thus results in a large number of tracks, among which the storms of interest need to be identified.

Identifying the storms in ERA-Interim is straightforward, because the selection of severe storms is based on the same dataset. For each of the 25 storms, the reference time and position of minimum MSLP given by Roberts et al. (2014) are searched for in the tracks obtained from the algorithm. The closest track is unambiguously identified this way and matches the reference track, although differences may arise, particularly at the beginning and end. As shown by Neu et al. (2013), such differences are a common issue when comparing storm tracking algorithms, which usually agree well for the mature phase of deep cyclones but

differ during the phases of cyclogenesis and cyclolysis. In particular, the algorithm of Pinto et al. (2005) tends to identify the cyclones earlier than others. Neu et al. (2013) emphasize that there is no best way of tracking storms, because there is no single definition of extratropical cyclones. As the same algorithm is applied here to both ERA-Interim and the reforecasts, potential biases due to the tracking method would likely cancel out.

In the reforecast, identifying the storms is less straightforward even at short lead times and quickly becomes ambiguous,

because the tracks diverge from ERA-Interim when the lead time increases. In earlier studies, Froude et al. (2007a, b) applied strict criteria in the location, timing and duration of tracks to identify storms in forecasts. While such criteria may be required for statistical studies, they would reject too many ensemble members for the sample of storms considered here, in particular at long lead time, and thus would bias the results towards "good" members only. Instead, the track closest to ERA-Interim is identified in each ensemble member without arbitrary criteria, based on the great-circle distance averaged over a 24-h period.

Two methods are compared for the definition of the 24-h period. In the first method, the period is defined as the first 24-h overlap between the track in the ensemble member and in ERA-Interim. If the track is not present at the time of initialization, it is further constrained to start in the ensemble member within 48 h of its first occurrence in ERA-Interim. In the second method, the period is simply defined as the day of maximum intensity.

The two methods are illustrated for the 7-day reforecast of the storm that hit the British Isles on 28 October 1996 ("u19961028",

Table 1). The storm took its origin in Hurricane Lili, which reached Europe after crossing the North Atlantic and undergoing extratropical transition (Browning et al., 1998). With the first method, the identified tracks start from the same location, because the storm is present in the reforecast at the time of initialization (Figure 2a). They later diverge and only two of them reach Europe, whereas the others remain over the central North Atlantic. With the second method in contrast, the identified tracks all reach Europe, as expected from the identification on the day of maximum intensity (Figure 2b). However, they start





from different regions spreading from the western to the eastern North Atlantic. In particular, no single track takes its origin in Hurricane Lili, i.e. the two methods do not show any common track. Although this case of extratropical transition is unique among the selected storms, it illustrates the difficulty of identifying storms in the reforecast. The most relevant method depends on the aims of the analysis; the first method focusing on the dynamics of the storm and the second one on its impact. Both

methods are therefore used here.

## 2.3   Storm Severity Index

While the intensity of a storm is commonly measured with its minimum MSLP, its severity mostly depends on the strength of the wind gusts, which is also controlled by the pressure gradient at the synoptic scale and by additional factors at the mesoscale and turbulent scale. In particular, insured losses have been shown to scale with the third power of the strongest wind gusts.

Following Klawa and Ulbrich (2003) and Leckebusch et al. (2008), a Storm Severity Index (SSI) is therefore defined as

$$SSI = \left( \frac{v_{max}}{v_{98}} - 1 \right)^3 \tag{1}$$

if $v_{max} > v_{98}$ and $SSI = 0$ otherwise, with $v_{max}$ the daily maximum wind gust and $v_{98}$ its local 98th climatological percentile. The scaling with $v_{98}$ accounts for the local adaptation to wind gusts, whose impact on infrastructure is weaker in exposed areas such as coasts and montains than in the continental flatlands for the same absolute wind speed (Klawa and Ulbrich, 2003).

The climatology of wind gusts is computed separately for ERA-Interim and the reforecast but for the same period of interest mid-October–mid-March 1995/96–2014/15. The resulting values of $v_{98}$ are higher in the reforecast, likely due to the higher model resolution. In particular, wind gusts are abnormally high over the topography in the first 6-h output of the reforecast, which suggests a problem with the spin-up of the model. The first 6 h are thus omitted for computing both $v_{max}$ and $v_{98}$. Wind gusts are also subject to caution in ERA-Interim but are still preferred to the wind speed (used by Leckebusch et al., 2008),

because they represent maximum values over a certain time period rather than instantaneous values and thus better sample storms with a large displacement velocity.

The daily maximum gusts in ERA-Interim are shown in Figure 3a and the resulting SSI in Figure 3b for storm Lothar on 26 December 1999. The strongest gusts are found over the Bay of Biscay but the highest SSI is found over southern Germany due to the lower values of the local model climatology. The SSI is then averaged over central Europe (defined as 40°N–60°N and

10°W–30°E; corresponds to map shown in Figure 3) to give a single value for the total severity of the storm, which can then be compared with the reforecast. This method is equivalent to the area SSI defined by Leckebusch et al. (2008). It is preferred to including the SSI along the track of the storm only (event SSI in Leckebusch et al., 2008), because of the ambiguous identification of the tracks in the reforecast. Among the 25 investigated storms, Lothar exhibits the highest averaged SSI in ERA-Interim, followed by Klaus, Martin and Kyrill (Table 1). These four storms are responsible for the four highest insurance

losses during the period of interest (Roberts et al., 2014), which suggests that the averaged SSI in ERA-Interim is a relevant measure of the severity of storms. Inaccuracies are still expected and attributed to mesoscale features that are not resoved by ERA-Interim and by non-meteorological factors such as the density of population and the insured capital.





## 2.4 Extreme Forecast Index and Shift of Tails

Forecasting extreme events is a challenge in numerical weather prediction, because predicted extremes tend to underestimate the magnitude of actual events. Lalaurette (2003) therefore introduced the Extreme Forecast Index (EFI), which measures the extremeness of an ensemble forecast as compared to the model climate rather than to the observed climate. The original for-
mulation of the EFI was revised by Zsótér (2006), who included a weighting function to emphasize the tails of the distribution and obtained

$$EFI = \frac{2}{\pi} \int\limits_0^1 \frac{p - F_f(p)}{\sqrt{p(1-p)}} dp \qquad (2)$$

with $F_f(p)$ the proportion of ensemble members lying below the $p$ quantile of the model climate. The EFI quantifies the deviation of an ensemble forecast from its climatological distribution with a unitless number between -1 (all members reach
record-breaking low values) and +1 (record-breaking high values).

Zsótér (2006) also introduced the Shift of Tails (SOT) as an additional index that focuses even more on the tail of the distribution

$$SOT(p) = -\frac{Q_f(p) - Q_c(p_0)}{Q_c(p) - Q_c(p_0)} \qquad (3)$$

with $Q_f(p)$ and $Q_c(p)$ the $p$ quantiles of the ensemble forecast and of the model climate, respectively. The SOT indicates if a
fraction of the ensemble members predicts an extreme event, even if the rest of the members do not. Following Zsótér (2006), $p$ is taken as the 90th percentile, i.e. the top two members of the 11-member ensemble reforecast. As in the operational ECMWF configuration, $p_0$ is taken as the 99th percentile of the model climate, which is smoother than the 100th percentile (maximum) used by Zsótér (2006). A positive value of SOT thus means that at least two members predict an extreme event that belongs to the top percent of the model climate.

Both EFI and SOT are computed here for daily maximum wind gusts. For consistency with the SSI, the model climate is defined from the period mid-October to mid-March 1995/96–2014/15. This contrasts with the operational ECMWF configuration, where the model climate is defined for each forecast within a one-month window centred around the initialization time. As the focus is on winter storms here, a seasonal model climate is preferred to avoid storms to be considered as more or less extreme depending on when they occur during the season. A longer period is also preferred to improve the representation of
the 99th percentile of the model climate, as the length of the operational configuration has been validated for precipitation and temperature but not for wind gusts (Zsoter et al., 2015). Finally, as in the operational configuration, the model climate is computed separately at each lead time to compensate for any drift of the reforecast.

Figure 4 illustrates the EFI and SOT for the 6-day reforecast of storm Lothar. High values of EFI spread over a broad region from the Atlantic Ocean to eastern Europe and exhibit stripes further eastward (Figure 4a). Positive values of SOT also spread
over a similar, broad region but the highest values are more concentrated (Figure 4b). This is due to the stronger emphasis on the tail of the distribution based on 2 members in SOT rather than on the whole ensemble in EFI. A comparison with ERA-Interim in Figure 3a indicates a skill of both EFI and SOT in predicting the strong gusts over parts of France, Switzerland and



Germany. However, it also shows a discrepancy between high EFI or SOT and weaker gusts over other regions. This suggests a potential for warnings but with possible false alarms, as already noted by Lalaurette (2003). The use of EFI and SOT thus requires an appropriate balance between hit rate and false alarm rate (Petroliagis and Pinson, 2014; Boisserie et al., 2016).

## 3  Predictability of storm characteristics

### 3.1  Position and intensity

The predictability of the selected storms is first evaluated for the position and intensity obtained from the storm tracking algorithm. The storms are identified in the reforecast at the time of first occurence and compared with ERA-Interim at the time of maximum intensity. As the 10-day reforecasts are computed every Monday and Thursday, three lead times are available for most storms but only two for those which occurred on a Sunday. The average bias and spread are computed for each storm and lead time with the median and median absolute deviation, respectively, which are preferred to the mean and standard deviation to ensure robust statistics despite the small number of ensemble members.

On average over all storms, the predicted MSLP remains close to ERA-Interim until day 4, but exhibits a clear positive bias, i.e. it underestimates the intensity of storms from day 5 onwards (black curve in Figure 5a). The predicted MSLP also exhibits a large dispersion between the storms, which increases with increasing lead time (symbols in Figure 5a). The most striking outlier is storm Gero (red triangle), which shows the strongest positive biases with more than 60 and 40 hPa on days 5 and 8, respectively. Gero experienced an explosive cyclogenesis of 40 hPa in 24 h to reach 948 hPa on 11 January 2005, the deepest MSLP of the sample of storms (Table 1). This suggests an impact of the storm intensity on its predictability, although no systematic link is found in the sample of storms. For instance, the second and third deepest storms Oratia and Stephen, which also experienced an explosive cyclogenesis, show contrasting positive and negative biases in MSLP depending on the lead time (green triangle and blue circle in Figure 5a). The predicted MSLP of Gero also exhibits a negative bias on day 1, although this may be due to ERA-Interim underestimating the actual intensity due to its coarse horizontal resolution.

Concerning the position, the predicted longitude exhibits a negative bias on average, i.e. the storms are too slow in the reforecast from day 4 onwards (black curve in Figure 5b). A weak positive bias is present in the reforecast of the latitude but it does not appear to be significant (not shown). Similar to the predicted MSLP, the predicted longitude also exhibits a large dispersion between the storms, which increases with increasing lead time (symbols in Figure 5b). Storm Gero is again an outlier with strong negative biases at days 5 and 8 but the strongest biases are shown by ex-Lili at day 7 (blue square) and Dagmar at day 10 (blue cross). These two storms formed remotely from Europe, the former in the tropics (Browning et al., 1998, see also Figure 2) and the latter over the southeastern United States. This suggests a link between the poor predictability of the position and the difficulty at representing convective dynamics, especially during extratropical transition (e.g. Pantillon et al., 2013). However, storm "u19960207" shows a strong negative bias in longitude at day 7 (green square) though it developed over the eastern North Atlantic. This emphasizes that single factors can influence the predictability of specific storms but do not necessarily have a systematic impact.



As expected, the spread between the ensemble members increases regularly with the increasing lead time on average, both for the intensity (solid black curve in Figure 5c) and the position (solid black curve in Figure 5d). The spread is consistent with the median absolute error (dashed curve), which suggests that the ensemble reforecast is properly calibrated. However, a large dispersion is again found between the storms and the spread does not match the error for individual storms. The storms with a strong bias mentioned above tend to exhibit a small spread, i.e. their reforecast is overconfident. Inversely, other storms that have a small bias exhibit a large spread, i.e. their reforecast is overdispersive. For instance, the predicted MSLP of Joachim was very uncertain due to the sensitivity to the phasing of the storm with a Rossby wave train over the western North Atlantic (Lamberson et al., 2016, green crosses at days 7 and 10 on Figure 5c). The large uncertainty in the MSLP of Xynthia at day 3 (red plus) may be due to the sensitivity of its intensification to latent heat release during its unusual track over the subtropical North Atlantic (Ludwig et al., 2014) but this is not consistent with longer lead times. This again emphasizes the difficulty at pointing out a systematic link between physical factors and the predictability of storms.

### 3.2 Ensemble average and individual members

These results partly agree with findings of Froude et al. (2007a, b) from a systematic evaluation of the track of extratropical cyclones in earlier versions of the operational ECMWF ensemble forecast system. The motion of cyclones was also too slow in the forecast but their MSLP was too deep. Beyond the model version, these differences emphasize the dependency on the selection of cyclones. The underestimation of the intensity and speed of storms shown here may thus not be systematic in the ensemble reforecast but rather be related to the selection of deep cyclones that reach Europe. Froude et al. (2007b) further found a higher skill of the ensemble mean compared to the control forecast to predict the track and intensity of cyclones. Although not tested here, this result raises the question of the meaningfulness of the ensemble mean at lead times beyond a few days, when the identification of storms becomes ambiguous. In particular, the number of members still containing the storm decreases when the lead time increases, which biases the ensemble mean. In the extreme case of ex-Lili for instance, all members of the 10-day reforecast valid on the day of maximum intensity have lost track of the storm on the day it reaches Europe, making this metric meaningless.

Using the alternative identification method focusing on the day of maximum intensity ensures that a storm is identified in each member of the ensemble. The predictability can then be measured by the number of members that match the actual storm within certain thresholds in position and intensity. Using moderate thresholds of 10° great circle in distance and 10 hPa in MSLP bias (Figure 6a), the storms are captured by all 11 members on the first day of lead time only. This number then decreases and passes below the majority of members beyond day 5. Albeit arbitrary, the thresholds express reasonable criteria for the definition of the actual storm and roughly correspond to the median value of both bias and spread in position and intensity among all storms and lead times. Using more restrictive thresholds of 5° great circle in distance and 5 hPa in MSLP bias (Figure 6b), the storms are still captured by all 11 members at day 1 but are missed by the majority of members beyond day 3 already. These results suggest that the use of the ensemble mean to predict storms should be restricted to the first 2–4 days of lead time, although the exact limit depends on the thresholds and varies from storm to storm. The use of single members is discussed in the next Section.





## 3.3 Storm impact

The predictability of the selected storms is further evaluated for the impact of the wind gusts estimated from the SSI. Only the daily, spatially averaged SSI is evaluated here, without considering geographical information on where the storm occurred exactly. The reforecast is therefore evaluated for its ability to predict a severe storm on a specific day but anywhere over central

Europe. It is compared to ERA-Interim as a logarithmic difference, because the SSI is highly nonlinear (Equation 1) and spans several orders of magnitude between the least and the most severe storms of the selection (Table 1). Finally, although the SSI is scaled locally with separate model climates between the reforecast and ERA-Interim, the predicted distribution of SSI is overestimated overall. The overestimation is strongest for the low quantiles of the distribution then decreases to a factor of about 2 in the higher quantiles. The predicted SSI is thus divided by a factor of 2 for ease of comparison unless stated otherwise.

On average over all storms, the reforecast is close to ERA-Interim until day 3 but then drops by one order of magnitude and thus strongly underestimates the SSI at longer lead times (solid curve in Figure 7a). This drop is specific to the sample of severe storms and is not due to a systematic drift in the reforecast, which is illustrated by the 99th percentile of predicted SSI remaining almost constant with lead time (dashed curve). In addition, the average spread in SSI between ensemble members increases until day 3 only, before it decreases again when the average SSI drops (not shown). The reforecast is thus underdispersive at

longer lead time. As for the MSLP, however, the predicted SSI shows a large dispersion between the storms (symbols). For instance, the deep storms Gero and Oratia are again outliers with strong negative biases at days 5, 8 and 9, respectively, whereas a few other storms even exhibit a positive bias.

These results are confirmed by measuring the number of members that predict at least the SSI of ERA-Interim, which also drop at day 4 (Figure 7b). Note that this is a rather pessimistic estimation, as the predicted SSI is divided by a factor of 2.

Before the drop at day 4, the number of members is further separated into two groups with either a large majority or a small minority capturing the storms. This suggests that the reforecast systematically over- or underestimates the severity of individual storms. ERA-Interim may also contribute to the cases of overestimation by underestimating the actual SSI due to its limitation at representing the mesoscale structure of some storms. Beyond day 3, the reforecasts show a systematic underestimation of the SSI for almost all storms. However, at least one ensemble member on average still predicts the SSI of the storms until day

7, which suggests a potential for early warning based on individual members.

## 4 Skill for early warnings

### 4.1 Intense and extreme events

The results above show that even though the use of the ensemble average is restricted to the first 3 days of lead time, single members are able to predict the storms up to one week in advance or even beyond, as was already mentioned by Froude et al.

(2007b). However, these results are biased by the focus on the prediction of observed events (hits) without considering events that are predicted but not observed (false alarms). In the following, the skill of the reforecast is investigated not only for the selected storms but for the whole mid-October–mid-March 1995/96–2014/15 dataset, in order to include days both with and





without storms. It is computed with the Brier Score (Brier, 1950), which measures the ability of the reforecast to predict if an event will occur or not.

The Brier Score can be split into reliability, resolution and uncertainty components (Murphy, 1973). The reliability component measures the ability of the forecast to predict the observed frequency of events. A perfect reliability can be achieved

with a climatological forecast and is thus not sufficient to be useful. In contrast, the resolution component measures the ability of the forecast to distinguish between events and non-events, which can not be achieved with a climatological forecast. The uncertainty component finally measures the sampling uncertainty inherent to the events. The Brier Score is further compared to a climatological forecast to obtain the Brier Skill Score (BSS), i.e. the actual skill of the reforecast, which is in turn split as

$$BSS = 1 - B_{rel} - B_{res} \tag{4}$$

into reliability and resolution components $B_{rel}$ and $B_{res}$ (e.g. Jolliffe and Stephenson, 2012).

The skill of the reforecast is first investigated for intense events defined as the top 5% of the SSI, which contain the 7–8 most severe storms per year on average. Percentiles are preferred to absolute values, because ERA-Interim and the reforecast exhibit different distributions of SSI. The frequency of intense events is then by definition the same (5%) in the reforecast than in ERA-Interim and thus the reliability component remains close to zero (perfect skill, Figure 8a). The non-zero value

reflects the sampling uncertainty. In contrast, the resolution component increases regularly with lead time to approach 1 (no skill). Therefore, the Brier Skill Score follows – with inversed sign – the evolution of the resolution component and decreases regularily until it vanishes (no skill) at day 9. The reforecast thus clearly exhibits positive skill, albeit small, at predicting intense events until day 8.

The skill is less clear for extreme events defined as the top 1% of the SSI. These contain the 30 most severe storms of the

whole dataset and approximately match the 25 selected storms in ERA-Interim. Surprisingly, the reforecast does not show any skill at day 1 (Figure 8b). This is linked to a high value of the resolution component (low skill) and may again be due to a problem with the spin-up of the model. The resolution component then regularily increases with increasing lead time as expected. In contrast, the reliability component shows an irregular evolution with lead time and large values reflecting a large sampling uncertainty. This emphasizes that the dataset is too limited to investigate extreme events, which on average represent

8.2 events per lead time only. As a result, the Brier Skill Score suggests that the reforecast exhibits some skill at predicting extreme events until day 6 but it suffers from the same irregular evolution with lead time.

## 4.2  Area covered by damaging gusts

The potential for early warnings of strong gusts is further investigated with the EFI and SOT, which are both designed for this purpose by highlighting the behaviour of the most extreme ensemble members. As noted by Lalaurette (2003) already, the EFI

gives useful warnings of extreme events but also frequent false alarms. Petroliagis and Pinson (2014) therefore suggested the use of an optimal threshold to balance between hits and false alarms, a higher (lower) threshold increasing (decreasing) both the hits and the false alarms. Boisserie et al. (2016) further suggest to maximize the Heidke Skill Score (Heidke, 1926) as a trade-off between hit rate and false alarm rate. Following these authors, an optimal threshold is looked for to predict gusts that





exceed the local 98th climatological percentile in ERA-Interim. This value is taken for consistency with the SSI. In contrast with the previous studies, however, which focused on specific storms or storm intensities, an optimal threshold is first computed for the whole dataset and only then applied to the selected storms. This ensures that the result is not biased by verifying the forecast with extreme events only.

As shown in Figure 9a, the optimal threshold in EFI decreases with lead time, because both hit rate and false alarm rate decrease with lead time for a given threshold. In contrast, the optimal threshold in SOT is stable until day 6 and decreases at longer lead times only (Figure 9b). This is due to the increase in false alarm rate with lead time for a given threshold in this case, which compensates for the decrease in hit rate (not shown). A constant threshold is thus suitable for the SOT and in the early range only. The dependency of the optimal thresholds on the lead time should else be taken into account for warnings.

The optimal thresholds further show seasonal and regional variability (not shown), which could also be included to improve warnings. For the sake of simplicity, however, they are not considered here.

Although the optimal threshold exhibits a different evolution with lead time between the EFI and the SOT, the corresponding Heidke Skill Score is very similar, with a slightly higher value for the EFI. It decreases regularly with increasing lead time but remains above zero (no skill) until day 10, the longest lead time investigated here. The decrease tightly follows the hit

rate, while the false alarm rate slowly increases but remains small due to the rarity of events by definition of the local 98th climatological percentile. Note that the false alarm rate, which is conditioned by the events that are not observed, should not be confused with the false alarm ratio, which is conditioned by the events that are not forecast. These results demonstrate the actual potential of both EFI and SOT for the early warning of strong gusts. If the local 99th climatological percentile is preferred to defined extreme events, as in early studies, the optimal thresholds need to be levelled up and the resulting skill

becomes lower but it also remains positive until day 10 (not shown).

### 4.3   Application to the selected storms

The optimal thresholds described above are applied to the EFI and SOT for the selected severe storms in the reforecast. The Heidke Skill Score is again used as a trade-off between hit rate and false alarm rate. It is computed for the prediction of gusts over the central European domain on the day of maximum intensity of each storm. As for the whole dataset, the EFI (Figure

10a) and the SOT (Figure 10b) exhibit similar Heidke Skill Score on average, which lies around 0.8 during the first two days (high skill) and then decreases with increasing lead time until vanishing at day 10 (no skill). In particular, before day 10, the Heidke Skill Score is higher for the storms (solid curves) than for the whole dataset (dashed curves). It is related to higher hit rates for the storms, which enhance the skill despite higher false alarm rates (not shown). This does not necessarily mean that the reforecast is more skillful at predicting the presence than the absence of storms but rather emphasizes how focusing on

observed events can bias the verification.

Beyond these average properties, the reforecasts of the storms exhibit contrasting skill from case to case. The dispersion between the storms quickly increases with increasing lead time and the Heidke Skill Score of some storms approaches zero or becomes negative from day 6 onwards (symbols on Figure 10). A poor skill is found in both EFI and SOT for storms Lili at day 7 and Gero at day 8 in association with a low hit rate, as well as for storm Joachim at day 7 in association with a high false





alarm rate. This is consistent with the large biases in MSLP and longitude and the large spread in MSLP, respectively, found for these storms. Other storms contrast between poor skill in EFI and good skill in SOT, as Yuma at day 4, which was remarked for its difficult forecast as it occurred (Young and Grahame, 1999), and Xynthia at day 6. The higher skill could be due to the high hit rate of the SOT compared to the EFI, as suggested by Boisserie et al. (2016). However, no difference is found here on

average in the whole sample.

    Storm Yuma has the lowest area of strong gusts of the whole dataset, followed by Lili, Gero and Xynthia (Table 1), which suggest a link between storm size and predictability. However, such a link is not systematic, as shown by storm Xaver, which exhibits almost no skill at day 6 in both EFI and SOT though one of the largest area of the dataset. Finally, storm Xynthia exhibits a surprisingly high skill at day 10 in both EFI and SOT thanks to a high hit rate. This constitutes an outlier compared to

all other storms, which show no skill at that lead time. However, none of the ensemble members predicts the actual development of Xynthia over the subtropical North Atlantic (Ludwig et al., 2014). Instead, several members predict a storm forming over the central North Atlantic but reaching the Iberian Peninsula on the same day as Xynthia. Although this successful reforecast could be due to chance rather than to the actual skill of the model, it illustrates how predicting individual storms becomes ambiguous at long range but suggests a potential for predicting an environment favorable to storm development.

**5  Conclusions**

The synoptic-scale predictability of 25 severe historical winter storms over central Europe is revisited by taking advantage of the ECMWF ensemble retrospective forecast (reforecast), which offers a homogeneous dataset over 20 years with a state-of-the-art ensemble prediction system. The winter 2015/16 model version is used here and contains 11 ensemble members with a horizontal grid spacing of 30 km up to 10 days lead time that are computed twice weekly for the mid-October–mid-March

1995/96–2014/15 period. The predictability of the storms is investigated with different metrics to include their dynamics, severity and spatial extension. A storm tracking algorithm delivers the position and intensity of the storms (Figure 1a), which are identified in the reforecast either at the time of first occurence or at the time of maximum intensity. The Storm Severity Index (SSI) estimates the actual impact of the storms (Figure 1b) using the strength of wind gusts exceeding the local 98th climatological percentile. The Extreme Forecast Index (EFI) and Shift of Tails (SOT) finally predict the area covered by strong

gusts (Figure 1c) by measuring the deviation of the ensemble forecast from the model climate. The metrics are combined to assess the reforecast against the ECMWF retrospective analysis (reanalysis) ERA-Interim.

    The ensemble average is unbiased until day 3 to predict the position and minimum MSLP of the storms on the day of maximum intensity. At longer lead times, however, it systematically underestimates the speed of motion and the depth of the storms. This bias is accompanied by an increase in ensemble spread by a similar magnitude, which suggests that the ensemble

is calibrated, but only a minority of ensemble members still captures the actual storm at lead times beyond 3–5 days. This questions the relevance of using the ensemble average at longer lead times. This differs from a classical situation of averaging the ensemble members to smooth the unresolved scales, as the variables of interests are objects here rather than continuous fields. The ensemble average further underestimates the SSI of the storms at lead times longer than 3 days and the relative




error reaches several orders of magnitude. The ensemble spread drops by orders of magnitude, which shows that the SSI of the storms is systematically underestimated. In contrast, there is no general drift in the reforecast, where severe events are present up to 10 days. Similarly, the biases in position and intensity may not be systematic but rather be due to the focus on intense storms that reach Europe. These results suggest that relevant predictions of storm properties are restricted to the first 2–4 days

of lead time. This suggestion is supported by the ambiguousness at identifying the storms at longer lead times in the reforecast.

A different methodology is therefore required at lead times longer than the 2–4 days horizon. Although they are missed by the ensemble average, the position, intensity and severity of the storms are captured by some members up to one week in advance or even beyond. As suggested by earlier studies, the whole distribution of the ensemble should thus be used by shifting the focus from the average and spread to individual members for the prediction of extreme events. The danger with

this approach, however, is to verify the predictions with regard to observed events only, i.e. by concentrating on the hit rate without accounting for the false alarms. The predictability is therefore investigated here in the whole dataset of 20 winter seasons including both stormy and non-stormy days. Tracking is not used, because it can be applied to cyclones only and becomes ambiguous at long lead times. For intense events defined as the top 5% of the SSI, which span on average 7–8 days per winter, the reforecast exhibits a positive Brier Skill Score that regularly decreases until vanishing at day 9. For extreme

events defined as the top 1% of the SSI, which approximately correspond to the 25 historical storms, the reforecast appears to exhibit a similar skill but suffers from a large sampling uncertainty at longer lead time. The EFI and SOT indices confirm the skill of the reforecast at predicting the area covered by strong wind gusts until day 10 for storms as for the whole dataset. These results highlight the potential for early warnings of storms but also the difficulty at verifying the forecast of extreme events, even with the extended dataset used here.

While the metrics agree on average, they exhibit a high case-to-case variability. The predictability is particularly low for a few storms involving an explosive cyclogenesis, a tropical origin or a small area. However, no systematic pattern is found among the sample of storms and their predictability partly lacks consistency between lead times. A possible explanation lies in the paucity of data at single lead times and for each storm, as the reforecast is computed twice a week and contains 11 members only. A more frequent initialisation and a larger amount of members may thus prove better ability at identifying systematic

links between the dynamics and predictability of storms. The NOAA ensemble reforecast offers a daily initialization (Hamill et al., 2013) but appears not to perform as well as its ECMWF counterpart for predicting wind over central Europe (Dabernig et al., 2015). Furthermore, even in the operational ECMWF ensemble forecast initialised every day and containing 50 members, Pirret et al. (2016) struggled to find a relation between the predictability and the intensity, track or physical processes of storms.

The predictability of the severe storms investigated here may not be linked to common factors but rather be due to charac-

teristics of the individual storms. This suggests a fundamental limitation due to the nature of severe storms, which are extreme events and often do not follow standard patterns. More case studies are thus needed to better understand the predictability of specific storm features at different scales. They should be eased by the new generation of global and regional reanalyses that become available with a high horizontal resolution able to better represent the storms. Alternatively, the focus of the predictability could be shifted from the storms to the large-scale conditions that favour their development (e.g. Pinto et al., 2014),

in particular at longer lead times, when the identification of storms is ambiguous among the ensemble members.



*Author contributions.* Florian Pantillon, Peter Knippertz and Ulrich Corsmeier defined the scientific scope of the study. Florian Pantillon performed the data analysis and wrote the paper. All authors discussed the results and commented on the paper.

*Acknowledgements.* Ensemble reforecast and ERA-Interim data provided courtesy ECMWF. The authors thank Philippe Arbogast, Dale Durran, Tim Hewson and Joaquim Pinto for discussions about the interpretation of the results. The research leading to these results has been done within the subproject C5 "Forecast uncertainty for peak surface gusts associated with European cold-season cyclones" of the Transregional Collaborative Research Center SFB / TRR 165 "Waves to Weather" funded by the German Research Foundation (DFG).



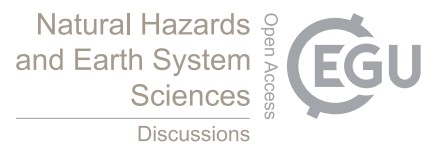

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




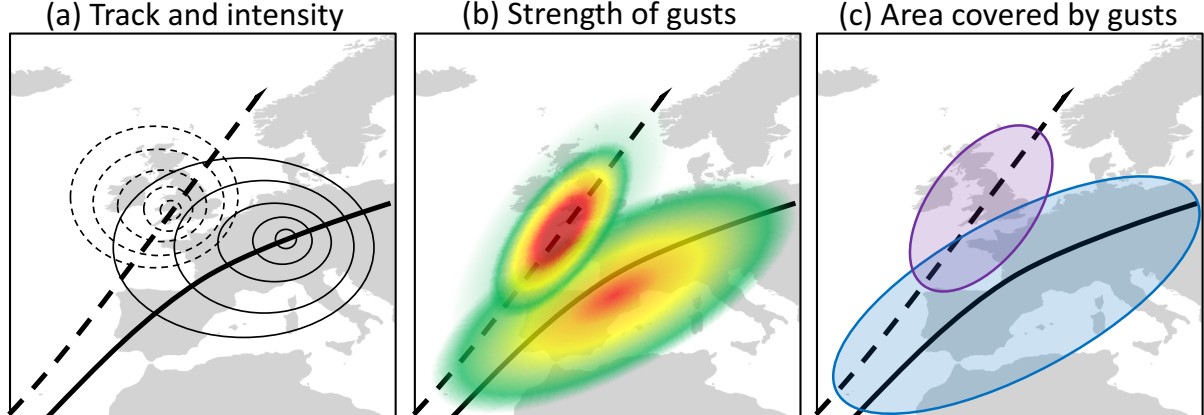

**Figure 1.** Schematic depiction of the three metrics used to evaluate the predictability of storms: based on the track and intensity of the storms (a), based on the strength of wind gusts (b) and based on the area covered by unusually strong gusts (c). See text for details.

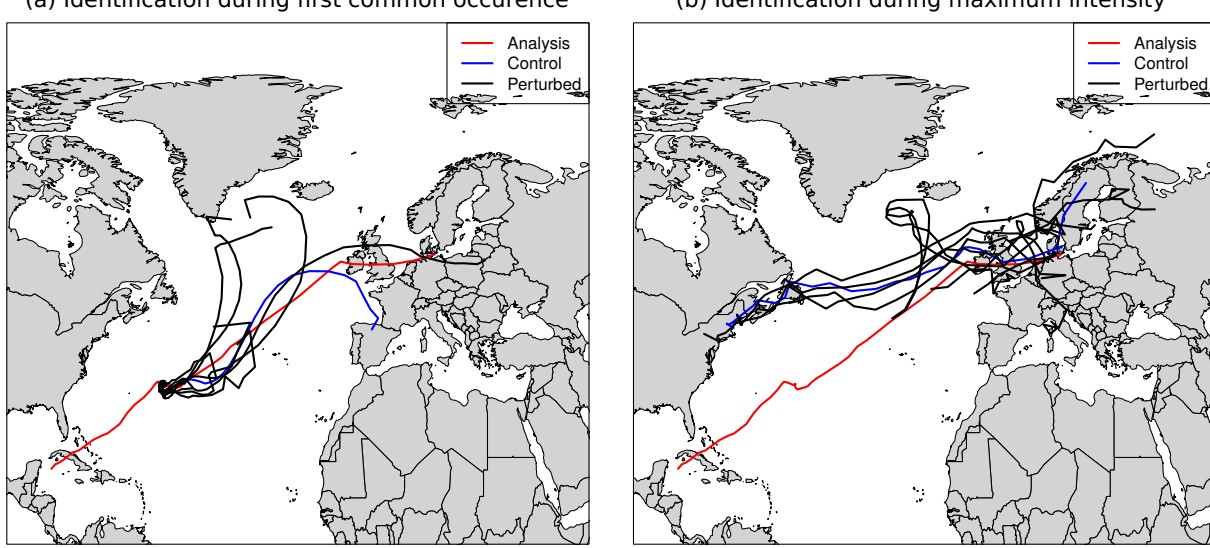

**Figure 2.** Example of the identified tracks of ex-hurricane Lili in the 6-day ensemble reforecast initialized on 22 October 1996 closest to ERA-Interim during the 24-h period of first common occurrence on 22 October (a) and of maximum intensity in ERA-Interim on 28 October (b).




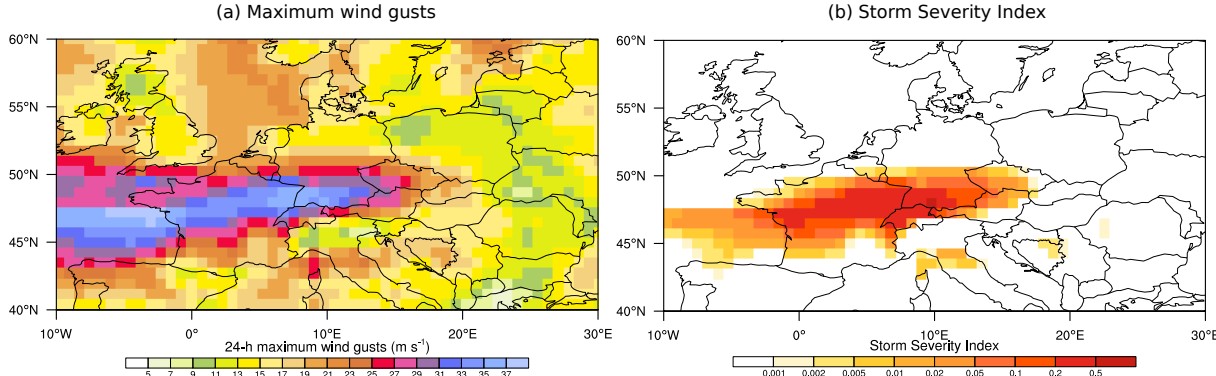

**Figure 3.** Example of the daily maximum wind gusts (a) and daily Storm Severity Index (b) for storm Lothar in ERA-Interim on 26 December 1999.

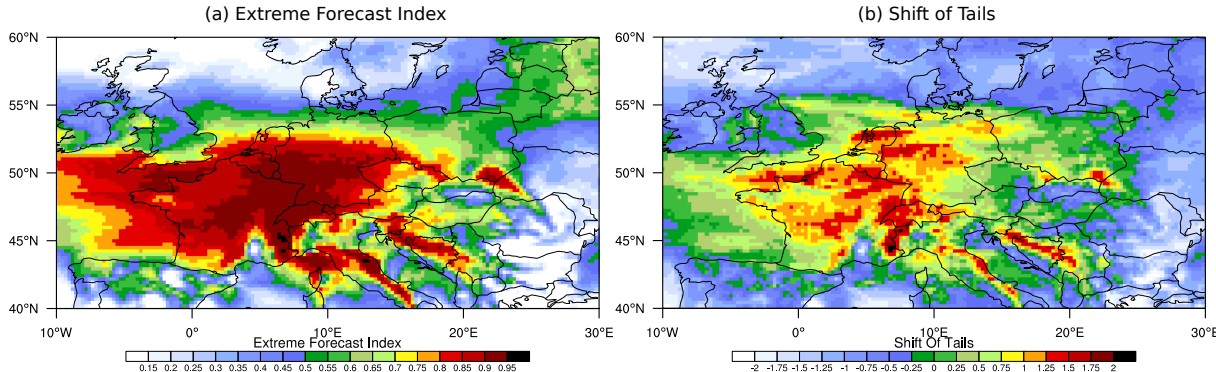

**Figure 4.** Example of the Extreme Forecast Index (a) and Shift of Tails (b) of daily maximum wind gusts for storm Lothar in the 6-day ensemble reforecast initialized on 21 December 1999 and valid on 26 December.





**Figure 5.** Position and intensity of the storms in the ensemble reforecast as identified at the time of first occurrence and compared on the day of maximum intensity: difference between the ensemble median and ERA-Interim (a, b) and median absolute deviation of the ensemble (c, d) in MSLP (a, c) and longitude (b, d). The symbols represent the storms as given in Table 1 and the solid black curve shows the median of the storms per lead time, while the dashed black curve in (c, d) further shows the median absolute error of the storms per lead time.



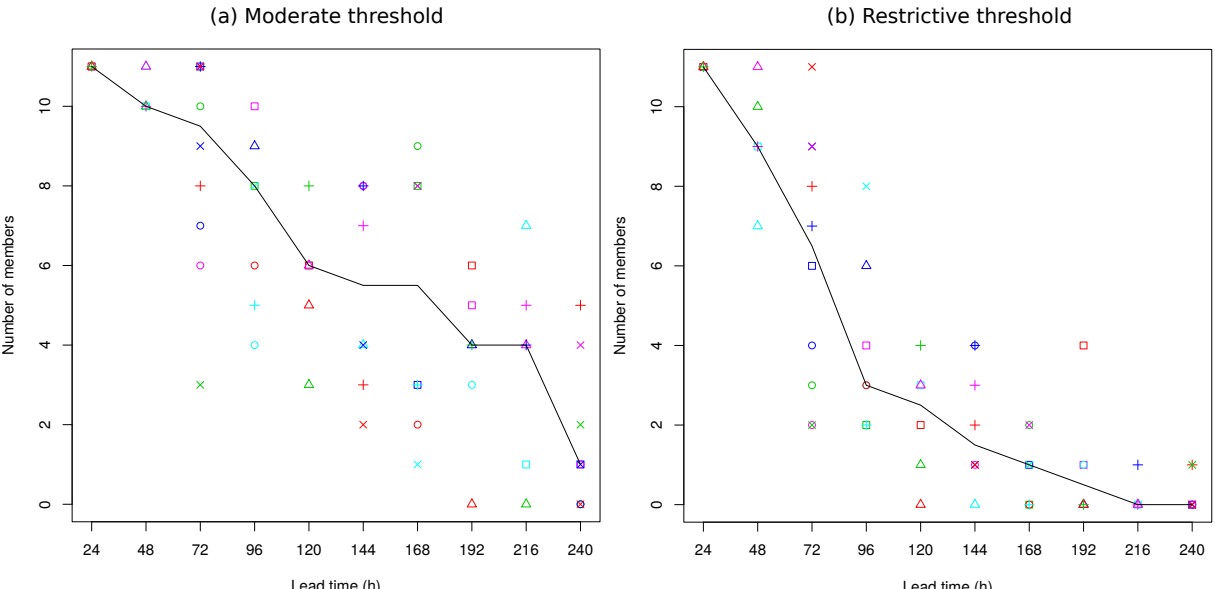

**Figure 6.** Position and intensity of the storms in the ensemble reforecast as identified and compared on the day of maximum intensity: number of ensemble members predicting the storm within 10 hPa and 10° great circle (a) or 5 hPa and 5° great circle (b) as compared to ERA-Interim in minimum MSLP and position, respectively. The symbols represent the storms as given in Table 1 and the black curve shows the median of the storms per lead time.



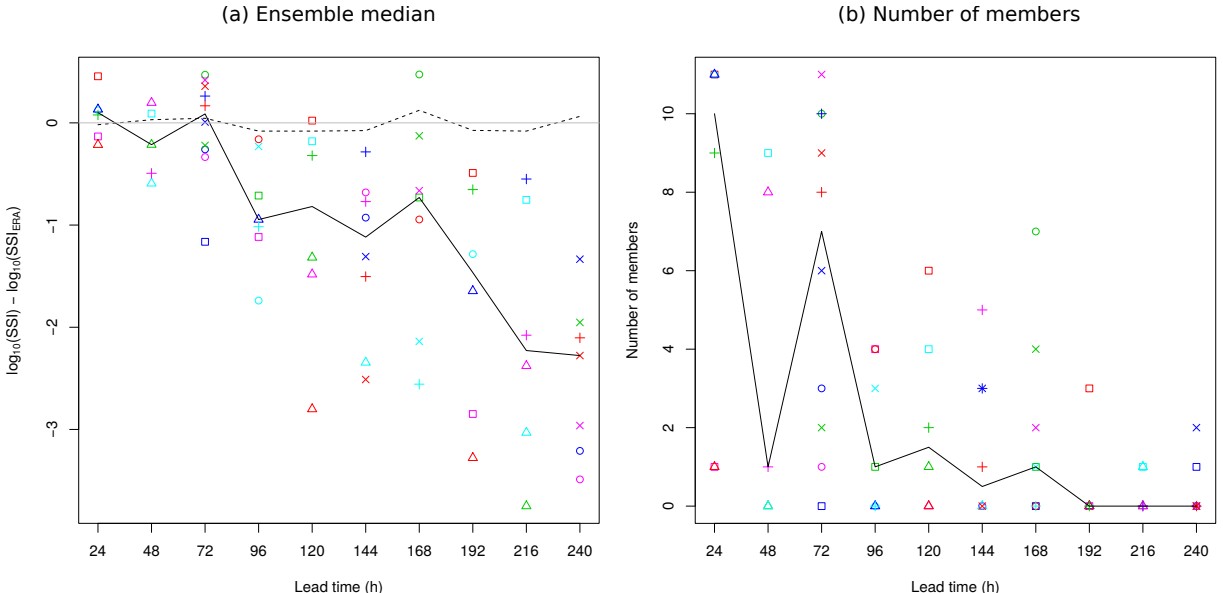

**Figure 7.** Severity of the storms in the ensemble reforecast on the day of maximum intensity: ensemble median of SSI as logarithmic difference with ERA-Interim (a) and number of members reaching the SSI of ERA-Interim (b). The predicted SSI is divided by a factor of 2 for ease of comparison. The symbols represent the storms as given in Table 1 and the solid black curve shows the median of the storms per lead time, while the dashed black curve in (a) further shows the 99th percentile of SSI compared between reforecast and ERA-Interim.

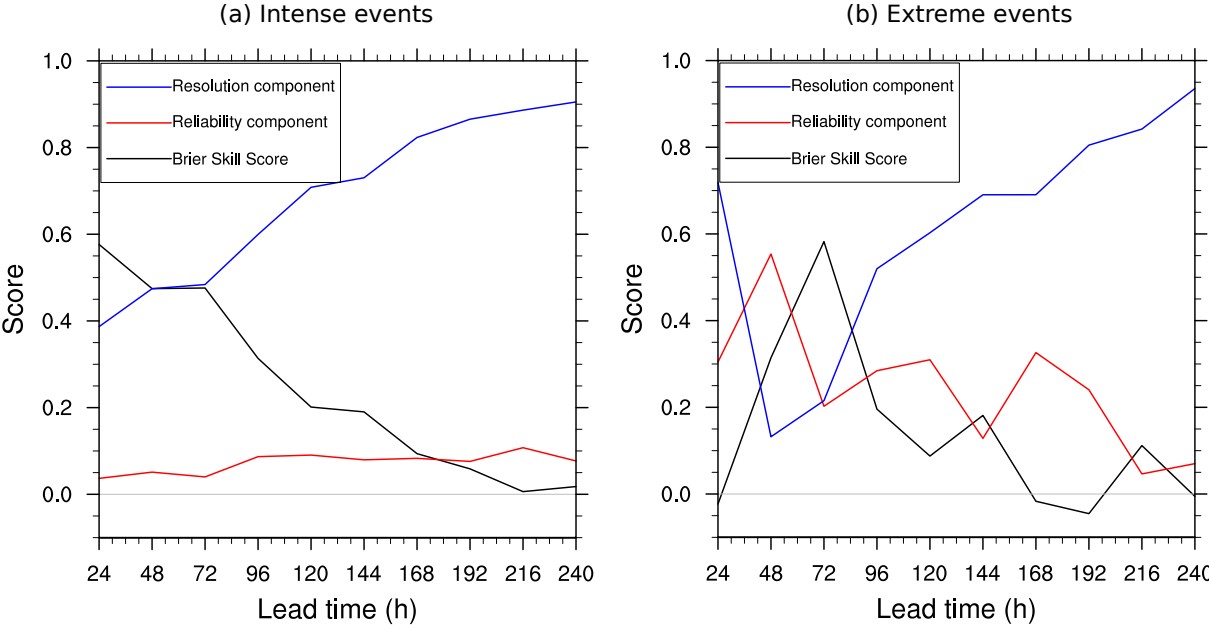

**Figure 8.** Brier Skill Score as a function of lead time for the SSI exceeding the 95th (a) and 99th percentiles of the model climatology (b). The Brier Skill Score is decomposed into resolution and reliability components (see Equation 4).





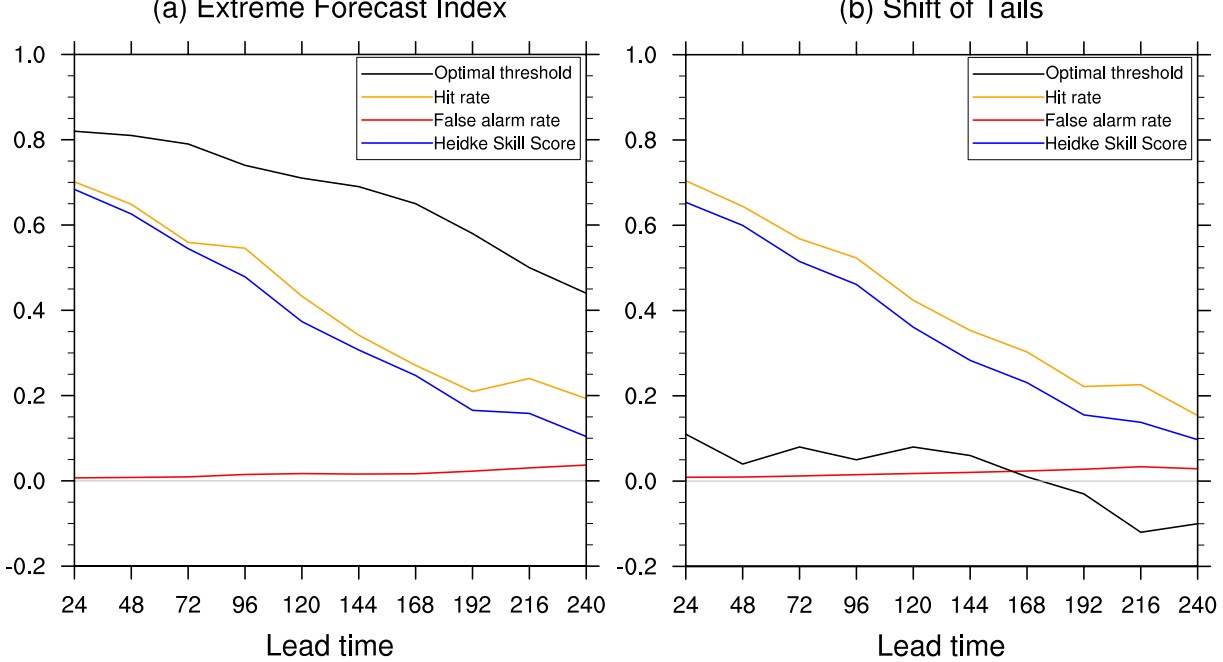

**Figure 9.** Optimal threshold and corresponding hit rate, false alarm rate and Heidke Skill Score for the Extreme Forecast Index (a) and the Shift of Tails (b) to predict gusts exceeding the local 98th percentile in ERA-Interim.

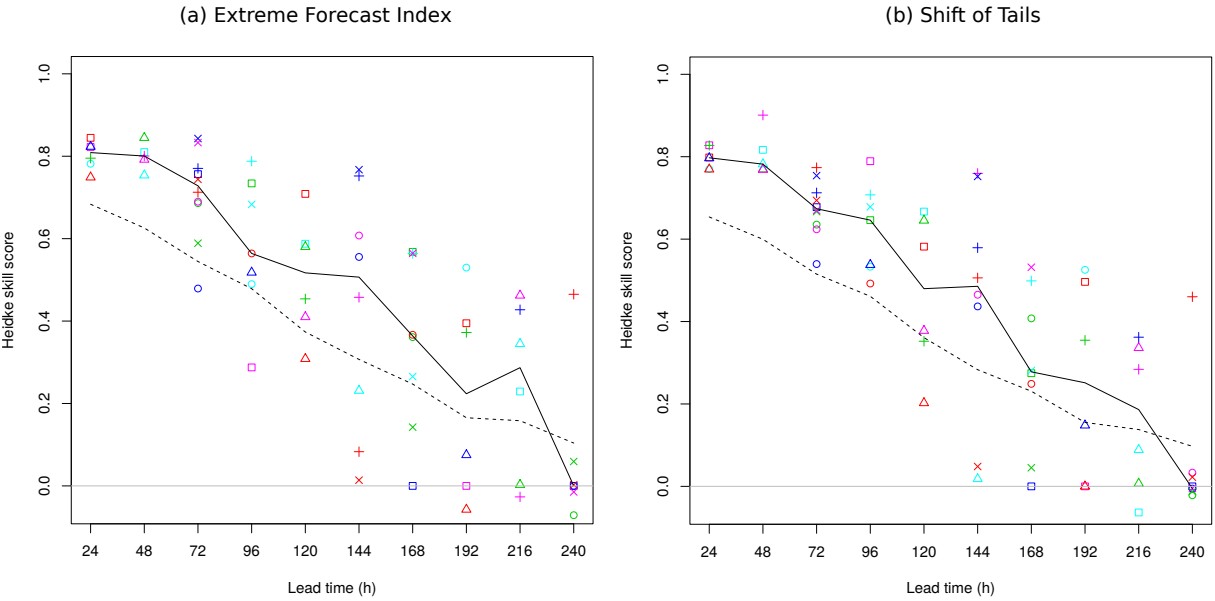

**Figure 10.** Heidke Skill Score for predicting gusts exceeding the local 98th climatological percentile of ERA-Interim using the Extreme Forecast Index (a) and the Shift Of Tails (b). The symbols represent the storms as given in Table 1 and the black curve shows the median of the storms per lead time, while the dashed curves illustrate the whole dataset for reference as in Figure 9.



**Table 1.** Chronological list of the 25 investigated storms with their characteristics in ERA-Interim on the day of maximum intensity: minimum Mean Sea Level Pressure (MSLP), Storm Severity Index (SSI) and area of central Europe covered by gusts exceeding the local 98th percentile. Some particularly high or low values are emphasized in bold.

| Symbol | Name | Date | MSLP (hPa) | SSI ($\times 10^{-3}$) | Area (%) |
|:---:|:---:|:---:|:---:|:---:|:---:|
| □ | u19960207 | 07 Feb 1996 | 976 | 3.0 | 11.1 |
| □ | u19961028 (ex-Lili) | 28 Oct 1996 | 970 | 0.40 | **7.3** |
| □ | u19961106 | 06 Nov 1996 | 960 | 0.48 | 20.8 |
| □ | Yuma | 24 Dec 1997 | 974 | 0.35 | **5.8** |
| □ | Fanny | 04 Jan 1998 | 966 | 2.0 | 16.6 |
| ○ | Xylia | 28 Oct 1998 | 966 | 0.64 | 28.3 |
| ○ | Stephen | 26 Dec 1998 | **950** | 2.4 | 21.0 |
| ○ | Anatol | 03 Dec 1999 | 956 | 5.1 | 28.7 |
| ○ | Lothar | 26 Dec 1999 | 976 | **15** | 23.7 |
| ○ | Martin | 27 Dec 1999 | 969 | **9.7** | 20.5 |
| △ | Oratia (Tora) | 30 Oct 2000 | **949** | 2.8 | 24.8 |
| △ | Jennifer | 28 Jan 2002 | 956 | 1.7 | 28.1 |
| △ | Jeanette | 27 Oct 2002 | 975 | 3.8 | 26.1 |
| △ | Erwin (Gudrun) | 08 Jan 2005 | 961 | 6.4 | 33.0 |
| △ | Gero | 11 Jan 2005 | **948** | 1.9 | **7.9** |
| + | Kyrill | 18 Jan 2007 | 963 | **6.7** | 35.5 |
| + | Emma | 01 Mar 2008 | 960 | 2.4 | 34.7 |
| + | Klaus | 24 Jan 2009 | 966 | **13** | 12.8 |
| + | Quinten | 09 Feb 2009 | 976 | 0.59 | 9.2 |
| + | Xynthia | 27 Feb 2010 | 968 | 2.7 | **8.7** |
| × | Joachim | 16 Dec 2011 | 966 | 3.5 | 31.0 |
| × | Dagmar (Patrick) | 26 Dec 2011 | 965 | 0.35 | 10.1 |
| × | Ulli | 03 Jan 2012 | 955 | 1.6 | 27.7 |
| × | Christian (St Jude) | 28 Oct 2013 | 969 | 0.91 | 18.7 |
| × | Xaver | 05 Dec 2013 | 962 | 2.3 | 34.9 |