# Peer review of "Revisiting the synoptic-scale predictability of severe European winter storms using ECMWF ensemble reforecasts"

_Natural Hazards and Earth System Sciences, 2017_

## Referee Comment (RC1) · Anonymous Referee #1 · 6 May 2017

Summary

The study assesses the predictability of severe storms over Europe in the most important season winter using the ECMWF ensemble forecasts. The authors concentrate on 25 events in the period 1995 to 2015 applying different metrics finding that these high impact events are predicted with skill up to 4 days. They also find skill for the area covered by these extreme events up to 10 days which may provide early warning opportunities. Still, the limited sample of only 25 storms shows strong inter-case variability. The small sample is a clear drawback of this study as it limits the reliability of the deduced skills and the author tend to overemphasize the results. Still the manuscript is nicely written and well structured. It certainly contains new findings,

which are fruitful for how to identify predictive skill for extreme events, so I certainly see that the manuscript is suitable for NHESS, if my minor to major comments are treated seriously.

Comments

P1,L9: Please change to 'potential for an early warning'.

P2,L1-7: You may add the study of Stucki et al. (2014, Nat. Hazards Earth Syst. Sci.) here.

P2,L29: Please change 'manuscript' to 'study'. P3,L15-16: As wind gusts are an important metric used in this study, you need to explain how this is derived in the reforecasts and how these gusts compare to observations.

P3,L24-25: How do the selected European wind storms compare to the storm catalogue provided by Stucki et al. (2014, Nat. Hazards Earth Syst. Sci.).

P4,L13: It would be nice to include the publication by Raible et al. (2008) who were the first to inter-compare cyclone tracking methods.

P4,L16: Please change to 'Neu et al. (2013) emphasized...'

P5,L11: It remains unclear which level is used for the wind – is it 10-m wind? Another question is whether the authors use wind gusts as v_max or sustained wind. If the authors use wind gusts they need to include a discussion on the parametrization used.

P5,L17-18: This could also a problem of the wind gust parameterization and not just a problem of the spin-up of the model. Stucki et al. (2016, Tellus) showed this how different gust parameterizations work over complex terrain showing strong changes from one to another parameterization.

P6, bottom line: This is why it is so important to say something about the gust parameterization and why the authors shall be encouraged to compare their result to direct observations also on areas with complex terrain.

[Figure]

P7,L27-29: If I understand the results correctly you only have two cases so such a strong statement that poor predictability is linked to process of extra tropical transition and convective dynamics cannot be derived, so the authors need to weak this statement and elsewhere in the manuscript.

P8,L34: It seems to be a bit awkward that the authors argue a high storm to storm dependency as in the rest of the paper they use all the cases to get some robust conclusion about predictability of severe storms which implies averaging over as much cases as possible, also the dependency to the threshold is expected as it is a matter of statistics that there is dependency to thresholds.

P10,L32-33: Change to ' further suggested to maximize . . . optimal threshold is used to predict gusts'

P11,L5: From Figure 9 I think that the hit rate decrease but the false alarm rate increase, correct?

P11-12, section 4.3: Well single storms are always special so I do not see why there is a need for this section.

P12,L16-26: Please shorten this part – it is a summary and not a conclusion.

P13,L8: Please cite the earlier studies and change 'should' to 'shall'.

P13,L20: I think the cases to case variability is expected.

P13,L21: The conclusion on low predictability for storms of tropical origin only relys on 2 cases so weaken this statement here.

References: Please get rid of the numerous errors in the reference list – this is annoying!

Figs. 5, 6, 7 and 10 needs to have increase axis labels as e.g. Fig. 8 has.

---

## Referee Comment (RC2) · Anonymous Referee #2 · 27 Jun 2017

The manuscript investigates the forecast skill of extreme storms (often called windstorms) in the ECMWF 20-year reforecasts. The ECMWF 20-year reforecasts are found to be skillful and ensemble spread well calibrated up to lead times of 3-5 days. After this the skill drops; storms are found to move too slowly and do not capture the intensity of observed events as measure by a Storm Severity Index. No systematic links between storm properties (size, intensity, etc..) and forecast skill is found. Some skill beyond 3-5 days is found using EFI and SOT indices, suggesting some utility for windstorm warnings at these lead times.

The paper will be of interest to weather forecasting community as it contains some

new and interesting results. In general, the paper is clear in its approach and figures are clear. I have a couple of specific comments on the paper (below) which should be addressed. I'd consider these major revisions, although I don't think it would take much to address these comments. Subsequently, I'd recommend the paper for publication provided these comments are fully addressed.

Specific Comments

1. Novelty of the study: The paper seems incremental in terms of progressing this area of science, since a lot of what is said in this paper was covered by Froude et al (2007). It would be helpful in terms of highlighting the novelty of this particular study if a) the Froude et al 2007 paper is discussed in the introduction and b) that the novelty of this paper is discussed in the conclusions.

2. Page 8. Lines 19 to 33 and figure 6. Figure 6 is very useful as it gives another sense of the utility of the reforecasts. However, I don't agree with some of the statements here about the validity or not using an ensemble mean. The statements seem rather confused to me. For example, we could say that for your MSLP analysis in figure 3 we shall choose a threshold error of 10hPa to indicate a useful forecast, and therefore we shouldn't compute an ensemble mean for when the bias in the ensemble mean went above this. You'd agree that this would sound like a strange and arbitrary thing to do, but this is effectively what you're arguing in this piece of text. This strange argument should be removed. Furthermore, I find it difficult to see how your results make the results of Froude et al 2007 invalid (line 19) as they looked at a different dataset. Could you be clearer here what you mean?

3. Page 12. Line 9 and Figure 7a and 7b. "The predicted SSI is thus divided by a factor of 2 for ease of comparison unless stated otherwise" Have you done this for the plots in Figure 7? If so then you will need to redo the plots without this adjustment and revise the text. There's no justification for dividing one dataset by an arbitrary number to make it more comparable to the other. Furthermore, why are the SSI much

larger in the reforecasts compared to ERA-I? Further down the page you say (Line 22), "ERA-Interim may also contribute to the cases of overestimation by underestimating the actual SSI due to its limitation at representing the mesoscale structure of some storms." You will need to provide some evidence of this statement (e.g. a reference). How much is ERA-I underestimating the true SSI? If ERA-I is very wrong, why are you using it as your main evaluation dataset? You'll need to address these questions.

Technical Comments

Line 5. ". . .storms are correctly predicted. . ." correctly would mean without any bias. Perhaps "well predicted" or "predicted with only small forecast errors" would be a better expression. Line 9. "However, a large variability is" should be "However, large variability is. . ." Line 10. "and does not appear. . .". What is it that does not appear? Do you mean the ". . .and the predictability of storms does not appear. . ."? Line 21 ". . .and of their forecast in numerical weather prediction systems." Perhaps could be better expressed as ". . .and of the ability of numerical weather prediction systems to forecast them." In addition, I don't disagree with the sentence but references need to be added.

Line 24-Line 28 and Figure 1. The sentences and the reference to Figure 1 do not belong in the introduction. They should be moved to the methods section.

Line 6. "In a second step, the mimima of MSLP are connected between subsequent model outputs every 6 h to form tracks, if their displacement velocity remains consistent in time." This second half of the sentence doesn't really make sense. Could you split the sentence and make clear what "their displacement velocity remains consistent in time" means? Line 8. "filtered to exclude storms with a weak Laplacian" Can you specify the threshold is?

Line 24. Do you include SSI values over ocean in your European spatial average? If so this doesn't seem like a good idea – does it make a difference if you use land-only values of SSI? Line 31 "resoved" should be "resolved"

Line 14 and line 24. "Dispersion" often has a very technical meaning. I think here what you mean is "variability". There are other examples of this in the manuscript that should be changed for readability.

Page 8.

Line 14. Rephrase "The motion of cyclones was also too slow in the forecast but their MSLP was too deep." As something like "The motion of cyclones was too slow in the forecasts. In addition, but the forecasted MSLP was too deep."

Line 4. "...on a specific day but anywhere over central..." would be better expressed as "...on a specific day over central..." Line 7 to 8. "...the predicted distribution of SSI is overestimated overall." I don't know what this means - what is the predicted distribution, is it the reforecasts? If so state this explicitly. Also state explicitly what the predicted distribution is overestimated relative to. Line 12. Can you add some detail to explain how you select events for the 99th percentile of SSI? Line 26. "Early Warning". This terms means something very different in different contexts. In some contexts, early warning only means 1-2 day lead time. I'd suggest being specific here in terms of timescale and call this section "Potential for Early Warnings on 5-10 day timescales".

Page 10.

Line 3-10. The description of Brier Skill Score should really be in the methods section.

Line 1. "This value is taken for consistency with the SSI." Can you say explicitly what this means? Line 19. "...the optimal thresholds need to be levelled up and..." What do you mean by levelled up? Do you mean increased?

Line 2. Should be "...which was noted..." Line 27 "The ensemble average is unbiased until day 3 to predict the position and minimum MSLP of the storms on the day of maximum intensity.", would be better expressed as, "The ensemble average has small biases until day 3 in terms of predicting the position and minimum MSLP of the storms on the day of maximum intensity." Line 30. Should be "...ensemble members captures the actual storm..." Line 30. "This bias is accompanied by an increase in ensemble spread by a similar magnitude, which suggests that the ensemble is calibrated, but only a minority of ensemble members still captures the actual storm at lead times beyond 3–5 days. This questions the relevance of using the ensemble average at longer lead times. This differs from a classical situation of averaging the ensemble members to smooth the unresolved scales, as the variables of interests are objects here rather than continuous fields" This appears to be a different argument than from earlier, where arbitrary thresholds were used to determine whether the ensemble contained the storm or not. Can you comment on this?

Line 17. "The EFI and SOT indices confirm the skill of the reforecast at predicting the area covered by strong wind gusts until day 10 for storms as for the whole dataset." You argued a few paragraphs ago that few of the ensemble members actually predicted storm beyond 3-5 days lead time. If that's the case, how can there be skill at the lead times of up to a week? This needs to be explained in the conclusions. Line 29. "The predictability of the severe storms investigated here may not be linked to common factors but rather be due to characteristics of the individual storms." You've just argued in the previous paragraph that you don't have enough data to make this statement! So

how can this statement also be true?

Figures

Table 1 "Some particularly high or low values are emphasized in bold." This is a confusing thing to do – either remove the bold numbers or decide on a sensible reason for using bold numbers.

Figure 3 and Figure 4. Font used in legends is too small and needs to be substantially larger to be readable.
* * *

---

## Author Comment (AC1) · 28 Jul 2017

**Summary**

*The study assesses the predictability of severe storms over Europe in the most impor-
tant season winter using the ECMWF ensemble forecasts. The authors concentrate
on 25 events in the period 1995 to 2015 applying different metrics finding that these
high impact events are predicted with skill up to 4 days. They also find skill for the
area covered by these extreme events up to 10 days which may provide early warn-
ing opportunities. Still, the limited sample of only 25 storms shows strong inter-case
variability. The small sample is a clear drawback of this study as it limits the reliabil-*

*ity of the deduced skills and the author tend to overemphasize the results. Still the
manuscript is nicely written and well structured. It certainly contains new findings,
which are fruitful for how to identify predictive skill for extreme events, so I certainly see
that the manuscript is suitable for NHESS, if my minor to major comments are treated
seriously.*

We thank the reviewer for his/her comments on the manuscript.

We will address all the comments below. In particular, we will clarify that we explore
the physical characteristics of some outliers that exhibit a particularly high or low pre-
dictability and avoid any suggestion of a systematic link between dynamics and pre-
dictability among the sample of storms. We will also detail and discuss the representa-
tion of wind gusts in the ensemble reforecast and in the reanalysis datasets. We hope
that these revisions will better support the results of the paper.

**Comments**

*P1,L9: Please change to 'potential for an early warning'.*

We prefer to change to "potential for early warnings".

*P2,L1-7: You may add the study of Stucki et al. (2014, Nat. Hazards Earth Syst. Sci.)
here.*

We will cite the Stucki et al. (2014) paper in the selection of storms, as stated below.

*P2,L29: Please change 'manuscript' to 'study'.*

We prefer to change "manuscript" to "paper".

*P3,L15-16: As wind gusts are an impor- tant metric used in this study, you need to
explain how this is derived in the reforecasts and how these gusts compare to obser-
vations.*

In both the ensemble reforecast and ERA-Interim, the wind gusts are computed from

the wind speed on the lowest model level and a turbulent component based on a similarity relation between the variability of the surface wind and the friction velocity. In the ensemble reforecast, which uses a more recent model version, the computation of wind gusts includes an additional component based on the low-level wind shear in convective situations. This additional component is expected to contribute to the strongest wind gusts when convection is embedded in the cold front. The resolution of the ensemble reforecast and ERA-Interim is known not to be sufficient to capture the strongest gusts due to mesoscale structures such as sting jets and to steep topography. However, as the focus here is on synoptic-scale aspects of winter storms, these limitations are likely rather unimportant. The comparison with ensemble forecasts remains fair, because their horizontal resolution is not sufficient to capture the strongest gusts either, and because the verification of wind gusts is based on values relative to the model climate rather than on absolute values.

We will add a paragraph in Section 2.1 to detail and discuss the representation of gusts in the model data.

*P3,L24-25: How do the selected European wind storms compare to the storm catalogue provided by Stucki et al. (2014, Nat. Hazards Earth Syst. Sci.).*

We will discuss the focus of the selection of storms on the United Kingdom due to a fixed threshold for wind gusts above 25 m s$^{-1}$, which is less often exceeded over continental Europe. We will further mention the Stucki et al. (2014) paper as another catalogue based on alternative criteria for the specific region of Switzerland.

*P4,L13: It would be nice to include the publication by Raible et al. (2008) who were the first to inter-compare cyclone tracking methods.*

The publication by Raible et al. (2008) will be included.

*P4,L16: Please change to 'Neu et al. (2013) emphasized. . .'*

We will implement the suggested change.

*P5,L11: It remains unclear which level is used for the wind – is it 10-m wind? Another question is whether the authors use wind gusts as $v_{m}ax$ or sustained wind. If the authors use wind gusts they need to include a discussion on the parametrization used.*

As stated above, we will add a paragraph in Section 2.1 to detail the representation of gusts in the model data.

*P5,L17-18: This could also a problem of the wind gust parameterization and not just a problem of the spin-up of the model. Stucki et al. (2016, Tellus) showed this how different gust parameterizations work over complex terrain showing strong changes from one to another parameterization.*

We will cite Stucki et al. (2016) in the discussion of Section 2.1 about the representation of gusts over complex terrain. However, the problem seems to be different here, as it occurs in the first 6-h output of the reforecast only and not during the subsequent outputs. This suggests that the problem is due to the model spin-up when the higher-resolution reforecast is initialized from the lower-resolution reanalysis. We will clarify this in the manuscript.

*P6, bottom line: This is why it is so important to say something about the gust parameterization and why the authors shall be encouraged to compare their result to direct observations also on areas with complex terrain.*

As stated above, we will add a paragraph in Section 2.1 to detail the representation of gusts in the model data.

*P7,L27-29: If I understand the results correctly you only have two cases so such a strong statement that poor predictability is linked to process of extra tropical transition and convective dynamics cannot be derived, so the authors need to weak this statement and elsewhere in the manuscript.*

We will clarify that the case of ex-Lili emphasizes the poor predictability of the position during extratropical transition due to the difficulty at representing convective dynamics

but that it is unique among the selected storms and that other cases that exhibit strong biases formed over very different regions, as e.g. Patrick over the southeastern United States and Jennifer (1996) over the eastern North Atlantic.

*P8,L34: It seems to be a bit awkward that the authors argue a high storm to storm dependency as in the rest of the paper they use all the cases to get some robust conclusion about predictability of severe storms which implies averaging over as much cases as possible, also the dependency to the threshold is expected as it is a matter of statistics that there is dependency to thresholds.*

We agree that the results depend on the exact thresholds but we believe that a limit of 2–4 days is realistic for the large majority of storms with a reasonable definition of a useful forecast of the actual storm for an operational forecaster. We will omit the storm-to-storm variability here, which is not necessary at this point of the discussion, and further revise and clarify the choice of thresholds.

*P10,L32-33: Change to ' further suggested to maximize . . . optimal threshold is used to predict gusts'*

We will change to "further suggested to maximize the Heidke Skill Score to define the optimal threshold".

*P11,L5: From Figure 9 I think that the hit rate decrease but the false alarm rate in-crease, correct?*

This is indeed confusing. We will clarify that this is due to a different balance between hit rate and false alarm rate.

*P11-12, section 4.3: Well single storms are always special so I do not see why there is a need for this section.*

The purpose of this section is double. Firstly, it illustrates how the verification of fore-casts can be biased by focusing on observed events only. Secondly, it explores possible links between the predictability of the storms and their physical characteristics.

We will clarify the separation between the characteristics of some outliers and the absence of a systematic link dynamics and predictability. We will further extend the discussion of the results and include a comparison with findings of previous studies.

*P12,L16-26: Please shorten this part – it is a summary and not a conclusion.*

We will shorten this part as suggested.

*P13,L8: Please cite the earlier studies and change 'should' to 'shall'.*

We will implement the suggested change and cite the earlier studies.

*P13,L20: I think the cases to case variability is expected.*

Again, we will clarify the separation between the characteristics of some outliers and the absence of a systematic link between dynamics and predictability. We will further discuss the limitation for the verification of extreme events and compare alternative methods.

*P13,L21: The conclusion on low predictability for storms of tropical origin only relys on 2 cases so weaken this statement here.*

We will clarify that the link with extratropical transition concerns a unique case in the dataset.

*References: Please get rid of the numerous errors in the reference list – this is annoy-ing!*

There appears to be a problem with the URL of several references. We will therefore omit the URL whenever the DOI is available.

*Figs. 5, 6, 7 and 10 needs to have increase axis labels as e.g. Fig. 8 has.*

We will increase the axis label in those figures as suggested.

2017-122, 2017.

---

## Author Comment (AC2) · 28 Jul 2017

*The manuscript investigates the forecast skill of extreme storms (often called windstorms) in the ECMWF 20-year reforecasts. The ECMWF 20-year reforecasts are found to be skillful and ensemble spread well calibrated up to lead times of 3-5 days. After this the skill drops; storms are found to move too slowly and do not capture the intensity of observed events as measure by a Storm Severity Index. No systematic links between storm properties (size, intensity, etc..) and forecast skill is found. Some skill beyond 3-5 days is found using EFI and SOT indices, suggesting some utility for windstorm warnings at these lead times.*

*The paper will be of interest to weather forecasting community as it contains some new and interesting results. In general, the paper is clear in its approach and figures are clear. I have a couple of specific comments on the paper (below) which should be addressed. I'd consider these major revisions, although I don't think it would take much to address these comments. Subsequently, I'd recommend the paper for publication provided these comments are fully addressed.*

We thank the reviewer for his/her comments on the manuscript.

We will address all the comments below. In particular, we will refer more to the results of earlier studies and emphasize the novelty of ours. We will also clarify the limitations of using the ensemble average for the track and intensity of the storms. We will finally discuss the representation of wind gusts in the ensemble reforecast and in the reanalysis datasets. We hope that these revisions will better support the results of the paper.

**Specific Comments**

*1. Novelty of the study: The paper seems incremental in terms of progressing this area of science, since a lot of what is said in this paper was covered by Froude et al (2007). It would be helpful in terms of highlighting the novelty of this particular study if a) the Froude et al 2007 paper is discussed in the introduction and b) that the novelty of this paper is discussed in the conclusions.*

There a two major differences between this paper and that of Froude: (1) Froude investigated extratropical cyclones in general, while this paper focuses on severe storms, which requires a much longer dataset to cover enough events; (2) Froude investigated the track and intensity only, while this paper uses two additional methods for the early warning and for the impact of storms, which both require forecasts of wind gusts.

We will clarify these two points by adding a paragraph in the introduction to discuss the papers of Froude and Pirret – using the same approach but applied to severe storms –

and by explicitly stating the novelty of the paper i.e. the combination of three different methods and the use of a long homogeneous dataset. We will additionally compare our results with those of these and other previous papers in the conclusions to emphasize the novelty of this paper.

*2. Page 8. Lines 19 to 33 and figure 6. Figure 6 is very useful as it gives another sense of the utility of the reforecasts. However, I don't agree with some of the statements here about the validity or not using an ensemble mean. The statements seem rather confused to me. For example, we could say that for your MSLP analysis in figure 3 we shall choose a threshold error of 10hPa to indicate a useful forecast, and therefore we shouldn't compute an ensemble mean for when the bias in the ensemble mean went above this. You'd agree that this would sound like a strange and arbitrary thing to do, but this is effectively what you're arguing in this piece of text. This strange argument should be removed. Furthermore, I find it difficult to see how your results make the results of Froude et al 2007 invalid (line 19) as they looked at a different dataset. Could you be clearer here what you mean?*

The use of the ensemble average is limited by two factors when the lead time increases: (1) the identification of the storms becomes ambiguous and (2) the number of members containing storms decreases. Both factors may bias the average towards tracks that are close to the analysis and thus overestimate the actual skill of the ensemble forecast. Thus an alternative metric is given as the number of members forecasting the "actual" storm. This obviously depends on how the "actual" storm is defined, but reasonable values suggest that the storms are predicted by almost all members (with high certainty) until day 2–4.

We will first clarify the limitations of the ensemble average due to the two factors mentioned above and then better justify the chosen thresholds with the alternative metric. However, we agree that the 2–4 day limit does not strictly restrict the range of utility of the ensemble average, as it depends on the exact threshold used, and will therefore remove this argument.

*3. Page 12. Line 9 and Figure 7a and 7b. "The predicted SSI is thus divided by a factor of 2 for ease of comparison unless stated otherwise" Have you done this for the plots in Figure 7? If so then you will need to redo the plots without this adjustment and revise the text. There's no justification for dividing one dataset by an arbitrary number to make it more comparable to the other. Furthermore, why are the SSI much larger in the reforecasts compared to ERA-I? Further down the page you say (Line 22), "ERA-Interim may also contribute to the cases of overestimation by underestimating the actual SSI due to its limitation at representing the mesoscale structure of some storms." You will need to provide some evidence of this statement (e.g. a reference). How much is ERA-I underestimating the true SSI? If ERA-I is very wrong, why are you using it as your main evaluation dataset? You'll need to address these questions.*

The SSI is systematically overestimated by a factor of 2 in the reforecast compared to ERA-Interim, not only for the selected storms but for intense and extreme events in general, as illustrated by the 95th and 99th percentiles of the whole reforecast dataset in Figure 7 (dotted and dashed curves). The overestimation is due to a longer tail of the distribution of wind gusts in the reforecast compared to ERA-Interim, which impacts the SSI although it is calibrated with a local climatological percentile (Equation 1). The overestimation must be accounted for when investigating the SSI of the selected storms; one means of doing this is by calibration of the reforecast by a factor of 2, as would likely be done in an operational context to correct a systematic bias.

However, we agree that the calibration might be confusing here. We will therefore present the results without calibration for the SSI of the storms. Instead, we will state that the overestimation until day 3 could be corrected, because it is systematic in the whole dataset, while the underestimation at longer lead times is specific to the storms and thus indicates a poor predictability. Finally, we will add a paragraph in the methods Section to discuss the representation of wind gusts in ERA-Interim and the reforecast datasets.

**Technical Comments**

*Page 1 Line 5. ". . .storms are correctly predicted. . ." correctly would mean without any bias. Perhaps "well predicted" or "predicted with only small forecast errors" would be a better expression.*

We will reword to "well predicted" as suggested.

*Line 9. "However, a large variability is" should be "However, large variability is. . ."*

We will correct this.

*Line 10. "and does not appear. . .". What is it that does not appear? Do you mean the ". . .and the predictability of storms does not appear. . ."?*

We will reword to "large variability is found between the individual storms and the predictability does not appear...".

*Line 21 ". . .and of their forecast in numerical weather prediction systems." Perhaps could be better expressed as ". . .and of the ability of numerical weather prediction systems to forecast them." In addition, I don't disagree with the sentence but references need to be added.*

We will clarify to "and on the ability of numerical weather prediction systems to forecast them, as detailed below".

*Page 2 Line 24-Line 28 and Figure 1. The sentences and the reference to Figure 1 do not belong in the introduction. They should be moved to the methods section.*

We will move the references to Figure 1 to the methods section as suggested.

*Page 4 Line 6. "In a second step, the mimima of MSLP are connected between subsequent model outputs every 6 h to form tracks, if their displacement velocity remains consistent in time." This second half of the sentence doesn't really make sense. Could you split the sentence and make clear what "their displacement velocity remains consistent in time" means?*

We will clarify to "the mimima of MSLP are connected between subsequent model outputs every 6 h, using a predicted velocity based on both the previous displacement and the steering by the environmental flow".

*Line 8. "filtered to exclude storms with a weak Laplacian" Can you specify the threshold is?*

We will specify "below 0.8 hPa $(^\circ$ great circle$)^{-2}$".

*Page 5 Line 24. Do you include SSI values over ocean in your European spatial average? If so this doesn't seem like a good idea – does it make a difference if you use land-only values of SSI?*

Indeed, SSI values are also included over adjacent ocean areas. This is to avoid large sensitivities to the predicted position of storms that track close to the coasts. In additon, although the impact of storms is expected over land mostly, including the ocean partially accounts for storm surges, which represent the main impact of some severe storms (e.g. Xynthia).

We will clarify this in the text.

*Line 31 "resoved" should be "resolved"*

We will correct this.

*Page 7 Line 14 and line 24. "Dispersion" often has a very technical meaning. I think here what you mean is "variability". There are other examples of this in the manuscript that should be changed for readability.*

We will replace "dispersion" by "variability" as suggested.

*Page 8. Line 14. Rephrase "The motion of cyclones was also too slow in the forecast but their MSLP was too deep." As something like "The motion of cyclones was too slow in the forecasts. In addition, but the forecasted MSLP was too deep."*

We will substiantially rewrite the paragraph to clarify the interpretation of the results.

*Page 9 Line 4. "...on a specific day but anywhere over central..." would be better expressed as "...on a specific day over central..."*

We will change this as suggested.

*Line 7 to 8. "...the predicted distribution of SSI is overestimated overall." I don't know what this means - what is the predicted distribution, is it the reforecasts? If so state this explicitly. Also state explicitly what the predicted distribution is overestimated relative to.*

We will rephrase to "although the SSI is scaled locally with separate model climates, it is systematically overestimated in the reforecast compared to ERA-Interim"

*Line 12. Can you add some detail to explain how you select events for the 99th percentile of SSI?*

We will precise "the 99th percentile of SSI values in the whole reforecast dataset".

*Line 26. "Early Warning". This terms means something very different in different contexts. In some contexts, early warning only means 1-2 day lead time. I'd suggest being specific here in terms of timescale and call this section "Potential for Early Warnings on 5-10 day timescales".*

We will rename the section as suggested.

*Page 10. Line 3-10. The description of Brier Skill Score should really be in the methods section.*

We will move the Brier Skill Score to the methods section as suggested.

*Page 11 Line 1. "This value is taken for consistency with the SSI." Can you say explicitly what this means?*

We will explain that "The 98th percentile represents the strength at which gusts become

damaging in the SSI (Equation 1)".

*Line 19. "...the optimal thresholds need to be levelled up and..." What do you mean by levelled up? Do you mean increased?*

Yes, we will change this.

*Page 12 Line 2. Should be "...which was noted..."*

We will change this as suggested.

*Line 27 "The ensemble average is unbiased until day 3 to predict the position and minimum MSLP of the storms on the day of maximum intensity.", would be better expressed as, "The ensemble average has small biases until day 3 in terms of predicting the position and minimum MSLP of the storms on the day of maximum intensity."*

We will change this as suggested.

*Line 30. Should be "...ensemble members captures the actual storm..."*

We will correct this as suggested.

*Line 30. "This bias is accompanied by an increase in ensemble spread by a similar magnitude, which suggests that the ensemble is calibrated, but only a minority of ensemble members still captures the actual storm at lead times beyond 3–5 days. This questions the relevance of using the ensemble average at longer lead times. This differs from a classical situation of averaging the ensemble members to smooth the unresolved scales, as the variables of interests are objects here rather than continuous fields" This appears to be a different argument than from earlier, where arbitrary thresholds were used to determine whether the ensemble contained the storm or not. Can you comment on this?*

We will clarify that there exists a limit of validity of the ensemble average for the track of the storms, because they are identified as objects, which are not always clearly defined. This contrast with the metrics based on the strength of wind gusts, which are

defined even in the absence of storm. We will further revise the paragraph based on the modifications in Section 3.2.

*Page 13 Line 17. "The EFI and SOT indices confirm the skill of the reforecast at predicting the area covered by strong wind gusts until day 10 for storms as for the whole dataset." You argued a few paragraphs ago that few of the ensemble members actually predicted storm beyond 3-5 days lead time. If that's the case, how can there be skill at the lead times of up to a week? This needs to be explained in the conclusions.*

We will clarify that the EFI and SOT, which emphasize the most extreme members, show a skill for predicting strong gusts until 9–10 days, while an accurate prediction of the position and intensity, which are based on the ensemble average, is limited to the first 2–4 days. We will further add a figure to clarify and summarize the results.

*Line 29. "The predictability of the severe storms investigated here may not be linked to common factors but rather be due to characteristics of the individual storms." You've just argued in the previous paragraph that you don't have enough data to make this statement! So how can this statement also be true?*

We agree that this statement is not justified and will clarify the limitation of the data for predicting extreme events.

*Figures Table 1 "Some particularly high or low values are emphasized in bold." This is a con- fusing thing to do – either remove the bold numbers or decide on a sensible reason for using bold numbers.*

We will specify that "The values corresponding to the deepest, most severe and small-est storms cited in the text are emphasized in bold".

*Figure 3 and Figure 4. Font used in legends is too small and needs to be substantially larger to be readable.*

We will increase the font size in legends as suggested.

---

## Author Response (AR2)

**Second response to Reviewer 1**

**Summary**

*Review of revised version of Revisiting the synoptic-scale predictability of severe European winter storms using ECMWF ensemble reforecasts*

*The authors almost included all my suggestions and comments in a reasonable way. Still as the manuscript underwent considerable changes some mainly technical corrections remain. Despite one point is still not treated adequately. The basis of the page and line notation below is the file nhess-2017-122-author_response-version1.pdf.*

We thank the reviewer for his/her careful reading of the revised version of the manuscript. A point-by-point answer is detailed below.

**Comments**

*P1,l11-14: This sentence is awkward, maybe the authors forgot to remove something when editing.*

The sentence has been split into two parts and now reads "However, large variability is found between the individual storms. The poor predictability of outliers appears related to their physical characteristics such as explosive intensification or small size."

*P1,l22-24: Concerning the statement '...question lies in the trends in frequency and intensity of winter storms in the current and future climate. This question is disputed due to little agreement between climate models and between identification methods (see Feser et al., 2015, for a review). However, the intensity of...' makes no sense. How cane the question be disputed by little agreement between climate models/methods? Also the 'However' makes a connection between the sentences which makes no sense, so please reformulate.*

This has been rephrased to "At longer time scales, a crucial question lies in the trends in frequency and intensity of winter storms in the current and future climate. To date, there is little agreement between climate models and between identification methods (see Feser et al., 2015, for a review). The intensity of storms..."

*P2,l25: Change throughout the text methodology to method as methodology is defined as is the systematic, theoretical analysis of the methods applied to a field of study.*

"Methodology" has been changed to "method" throughout the text.

*P2,l28: There is blank before a comma.*

This appears to be due to computing version changes with latexdiff. The blank is not present in the revised version (nhess-2017-122-manuscript-version2.pdf).

*P3,l13: Please cite the papers for the 'other authors'*

The "other authors" referred to those cited above. It has been removed for clarity.

*P4l25-27: If so why not comparing then max wind at the lowest level and remove the uncertainty source of gust parameterizations.*

The reason for this choice has been clarified to "Finally, modelled wind gusts well sample storms with a large displacement velocity, because they are computed as the maximum values over all model time steps between 6-hourly outputs. They are therefore preferred to modelled wind speeds, which are output as 6-hourly instantaneous values only."

*P6,l11: Please change to 'Neu et al. (2013) emphasized'*

It has been changed to "As first suggested by Raible et al. (2008) and later emphasized by Neu et al. (2013)".

*P7,l27 : Roberts et al. (2014) not in brackets.*

This has been corrected.

*P10, l20: The sentence is awkward, please clarify.*

The sentence has been simplified to "Finally, the large uncertainty in the MSLP of Xynthia at day 3 (red plus) may be due to the strong influence of latent heat release during the unusual track over the subtropical North Atlantic (Ludwig et al., 2014)."

*P12,L3: please change to 'which impacts the SSI'*

This has been changed.

*P12,L30-31: please change to 'they are forecasted'*

This has been corrected.

*P14, L8: please change to 'In contrast to the previous studies,' and add references*

This has been changed to "In contrast to the studies of Petroliagis and Pinson (2014) and Boisserie et al. (2016)"

*P16,L34: Methodology −> method*

"Methodology" has been changed to "method".

*P18,L16: Should −> can*

"Should" has been changed to "can".

*P18,L23: should −> shall*

"Should" has been changed to "shall".

*References: Still some titles are capitalized other not, use the style of the journal NHESS!*

This has been corrected.

*Figures: Still the labels use different font sizes when comparing it to Fig 8. (See old review)*

The font size of tick marks has been increased in Figures 5 and 10 to better match Figure 8. However, the figures are produced with different softwares and may thus still look different.

*Open issue from the old review There mentioned as P11-12, section 4.3:*

*The authors give an answer to this comment, but have not included the purpose of the section. So I suggest to write the purpose of this section explicitly at the beginning which will help the reader to follow the results.*

[revised manuscript text omitted]